# Acting in Delayed Environments with Non-Stationary Markov Policies

**Esther Derman**\*

Technion

estherderman@campus.technion.ac.il

**Gal Dalal**\*

Nvidia Research

gdalal@nvidia.com

**Shie Mannor**

Nvidia Research & Technion

shie@ee.technion.ac.il

## Abstract

The standard Markov Decision Process (MDP) formulation hinges on the assumption that an action is executed immediately after it was chosen. However, assuming it is often unrealistic and can lead to catastrophic failures in applications such as robotic manipulation, cloud computing, and finance. We introduce a framework for learning and planning in MDPs where the decision-maker commits actions that are executed with a delay of $m$ steps. The brute-force state augmentation baseline where the state is concatenated to the last $m$ committed actions suffers from an exponential complexity in $m$, as we show for policy iteration. We then prove that with execution delay, deterministic Markov policies in the original state-space are sufficient for attaining maximal reward, but need to be non-stationary. As for stationary Markov policies, we show they are sub-optimal in general. Consequently, we devise a non-stationary Q-learning style model-based algorithm that solves delayed execution tasks without resorting to state-augmentation. Experiments on tabular, physical, and Atari domains reveal that it converges quickly to high performance even for substantial delays, while standard approaches that either ignore the delay or rely on state-augmentation struggle or fail due to divergence. The code is available at https://github.com/galdl/rl_delay_basic.git.

## 1 Introduction

The body of work on reinforcement learning (RL) and planning problem setups has grown vast in recent decades. Examples for such distinctions are different objectives and constraints, assumptions on access to the model or logged trajectories, on-policy or off-policy paradigms, etc. (Puterman, 2014). However, the study of delay in RL remains scarce. It is almost always assumed the action is executed as soon as the agent chooses it. This assumption seldom holds in real-world applications (Dulac-Arnold et al., 2019). Latency in action execution can either stem from the increasing computational complexity of modern systems and related tasks, or the infrastructure itself. The wide range of such applications includes robotic manipulation, cloud computing, financial trading, sensor feedback in autonomous systems, and more. To elaborate, consider an autonomous vehicle required for immediate response to a sudden hazard on the highway. Driving at high speed, it suffers from perception module latency when inferring the surrounding scene, as well as delay in actuation once a decision has been made. While the latter phenomenon is an instance of *execution delay*, the former corresponds to *observation delay*. These two types of delay are in fact equivalent and can thus be treated with the same tools (Katsikopoulos & Engelbrecht, 2003).

**Related works.** The notion of delay is prominent in control theory with linear time-invariant systems (Bar-Ilan & Sulem, 1995; Dugard & Verriest, 1998; Richard, 2003; Fridman, 2014; Bruder & Pham, 2009). While the delayed control literature is vast, our work intersects with it mostly in motivation. In the above control theory formulations, the system evolves according to some known diffusion or stochastic differential equation. Differently, the discrete-time MDP framework does not require any structural assumption on the transition function or reward.

---

\*Equal contribution

A few works consider a delay in the reward signal rather than in observation or execution. Delayed reward has been studied on multi-armed bandits for deterministic and stochastic latencies (Joulani et al., 2013) and for the resulting arm credit assignment problem (Pike-Burke et al., 2017). In the MDP setting, Campbell et al. (2016) proposed a Q-learning variant for reward-delay that follows a Poisson distribution. Katsikopoulos & Engelbrecht (2003) considered three types of delay: observation, execution, and reward. Chen et al. (2020b) studied execution delay on multi-agent systems. The above works on MDPs employed state-augmentation with a primary focus on empirical evaluation of the degradation introduced by the delay. In this augmentation method, all missing information is concatenated with the original state to overcome the partial observability induced by the delay. The main drawback of this embedding method is the exponential growth of the state-space with the delay value (Walsh et al., 2009; Chen et al., 2020a) and, in the case of (Chen et al., 2020b), an additional growth that is polynomial with the number of agents.

Walsh et al. (2009) avoided state-augmentation in MDPs with delayed feedback via a planning approach. By assuming the transition kernel to be close to deterministic, their *model-based simulation* (MBS) algorithm relies on a most-likely present state estimate. Since the Delayed-Q algorithm we devise here resembles to MBS in spirit, we highlight crucial differences between them: First, MBS is a conceptual algorithm that requires the state-space to be finite or discretized. This makes it highly sensitive to the state-space size, as we shall demonstrate in Sec. 7 [Fig. 5(c)], prohibiting it from running on domains like Atari. Differently, Delayed-Q works with the original, possibly continuous state-space. Second, MBS is an offline algorithm: it estimates a surrogate, non-delayed MDP from samples, and only then does it solve that MDP to obtain the optimal policy (Walsh et al., 2009)[Alg. 2, l. 16]. This is inapplicable to large continuous domains and is again in contrast to Delayed-Q.

Recent studies considered a concurrent control setting where action sampling occurs simultaneously with state transition (Ramstedt & Pal, 2019; Xiao et al., 2020). Both assumed a single action selection between two consecutive observations, thus reducing the problem to an MDP with execution delay of $m = 1$. Chen et al. (2020a) have generalized it to an arbitrary number of actions between two observations. Hester & Stone (2013) addressed execution delay in the braking control of autonomous vehicles with a relatively low delay of $m \leqslant 3$. All these works employ state-augmentation to preserve the Markov property of the process, whereas we are interested whether this restriction can be lifted. Additionally, they studied policy-gradient (policy-based) methods, while we introduce a Q-learning style (value-based) algorithm. Likewise, Firoiu et al. (2018) proposed a modified version of the policy-based IMPALA (Espeholt et al., 2018) which is evaluated on a single video game with delay values of $m \leqslant 7$. To the best of our knowledge, our work is the first to tackle a delayed variant of the popular Atari suite (Bellemare et al., 2013).

**Contributions.** Revisiting RL with execution delay both in theory and practice, we introduce:

1. Analysis of a delayed MDP quantifying the trade-off between stochasticity and delay.

2. The first tight upper and lower complexity bounds on policy iteration for action-augmented MDPs. We stress that this is also a contribution to general RL theory of non-delayed MDPs.

3. A new formalism of execution-delay MDPs that avoids action-embedding. Using it, we prove that out of the larger set of history-dependent policies, restricting to non-stationary deterministic Markov policies is sufficient for optimality in delayed MDPs. We also derive a Bellman-type recursion for a *delayed value function*.

4. A model-based DQN-style algorithm that yields non-stationary Markov policies. Our algorithm outperforms the alternative standard and state-augmented DDQN in 39 of 42 experiments spanning over 3 environment categories and delay of up to $m = 25$.

## 2 PRELIMINARIES: NON-DELAYED STANDARD MDP

Here, we describe the standard non-delayed MDP setup. Later, in Sec. 5, we introduce its generalization to the delayed case. We follow and extend notations from (Puterman, 2014)[Sec. 2.1.]. An infinite horizon discounted MDP is a tuple $(\mathcal{S}, \mathcal{A}, P, r, \gamma)$ where $\mathcal{S}$ and $\mathcal{A}$ are finite state and action spaces, $P : \mathcal{S} \times \mathcal{A} \to \Delta_{\mathcal{S}}$ is a transition kernel, the reward $r : \mathcal{S} \times \mathcal{A} \to \mathbb{R}$ is a bounded function, and $\gamma \in [0, 1)$ is a discount factor. At time $t$, the agent is in $s_t$ and draws an action $a_t$ according to a decision rule $d_t$ that maps past information to a probability distribution $q_{d_t}$ over the action set. Once $a_t$ is taken, the agent receives a reward $r(s_t, a_t)$.

A decision rule can be history-dependent (H) or Markovian (M) , and randomized (R) or deterministic (D). Denote by $\mathcal{H}_t$ the set of possible histories up to time $t$. Then, a history-dependent decision-rule is given by $d_t : \mathcal{H}_t \to \Delta_\mathcal{A}$ with $h_t \mapsto q_{d_t(h_t)}(\cdot)$. A Markovian decision-rule, on the other hand, maps states to actions, i.e., $d_t : \mathcal{S} \to \Delta_\mathcal{A}$ with $s \mapsto q_{d_t(s)}(\cdot)$. A *policy* $\pi := (d_t)_{t \geqslant 0}$ is a sequence of decision rules whose type dictates that of the policy. It can be either Markovian deterministic ($\Pi^{MD}$) or randomized ($\Pi^{MR}$), history-dependent deterministic ($\Pi^{HD}$) or randomized ($\Pi^{HR}$). It is stationary if its decision rules do not depend on time, i. e., $d_t = d$ for all $t \geqslant 0$. This defines the smaller class of stationary policies: deterministic ($\Pi^{SD}$) and randomized ($\Pi^{SR}$). Note that stationary policies are inherently Markovian. Indeed, at time $t = 0$, $d : \mathcal{H}_0 \to \Delta_\mathcal{A}$ is state-dependent because $\mathcal{H}_0 = \mathcal{S}$. Since the policy is stationary, i. e., $d_t = d \ \forall t$, subsequent decision rules are also state-dependent, thus Markovian. This makes $\Pi^{HR}$ the most general set and $\Pi^{SD}$ the most specific.

We denote probability model by $\mathbb{P}_0^\pi$, where the subscript 0 stands for the delay value $m = 0$. The related random variables are denoted by $\tilde{s}_t \in \mathcal{S}, \tilde{a}_t \in \mathcal{A}$ and $\tilde{h}_t \in (\mathcal{S} \times \mathcal{A})^t \times \mathcal{S}$. The value function given policy $\pi \in \Pi^{HR}$ is defined as $v^\pi(s) = \mathbb{E}_0^\pi \left[ \sum_{t=0}^\infty \gamma^t r(\tilde{s}_t, \tilde{a}_t) \,\middle|\, \tilde{s}_0 = s \right]$, where the expectation is taken with respect to (w.r.t.) $\mathbb{P}_0^\pi(\cdot|\tilde{s}_0 = s)$. Let the optimal value function

$$v^*(s) := \max_{\pi \in \Pi^{HR}} v^\pi(s), \quad \forall s \in \mathcal{S}. \tag{1}$$

Our goal is to find a policy $\pi^*$ that yields $v^*$, and it is known that focusing on stationary deterministic policies $\pi \in \Pi^{SD}$ is sufficient for reaching the optimum in (1) (Puterman, 2014)[Thm. 6.2.10.].

## 3 MDPs with Delay: A Degradation Example

In an MDP with execution delay[1] $m$, any action chosen at time $t$ is executed at $t + m$. Therefore, at each step, the agent witnesses the current state and action being executed, but selects a new action that will be applied in a future state. We assume that $m$ decided actions are already awaiting execution at $t = 0$, so at any given time, the queue of pending actions is of constant length $m$. As we illustrate in the next example, having a delay generally comes at a price.

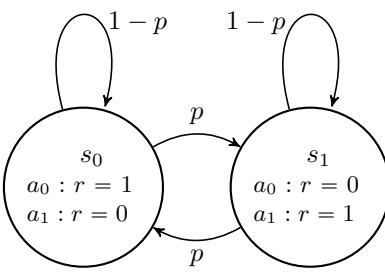

Figure 1: Degradation due to delay in a two-state MDP.

**Example 3.1** (Two-state MDP). *Consider the MDP in Fig. 1. It has two states and two actions: $\mathcal{S} = \{s_0, s_1\}, \mathcal{A} = \{a_0, a_1\}$. The transition kernel is independent of the action: for all $s, s' \in \mathcal{S}$ s.t. $s \neq s', P(s'|s, a) = P(s'|s) = p$ where $p \in [0.5, 1]$. The reward is positive for one of the two actions only: $r(s_0, a_0) = r(s_1, a_1) = 1, r(s_0, a_1) = r(s_1, a_0) = 0$.*

We inspect the return obtained from the commonly used set of stationary deterministic policies $\Pi^{SD}$. As expected, the highest possible return is attained when $m = 0$, but monotonically decreases with the delay, $m$, and increases with the level of certainty, $p$. We analytically quantify this effect in the following and give a proof in Appx. A.1.

**Proposition 3.1.** *For delay $m \in \mathbb{N}$ and $p \in [0.5, 1]$, the optimal return of $\pi^* \in \Pi^{SD}$ is $\frac{1 + (2p-1)^m}{2(1-\gamma)}$.*

**Remark 3.1.** *This result demonstrates a clear tradeoff between stochasticity and delay. For $p \to 0.5$ or $m \to \infty$, the return goes to its minimal value of $0.5/(1 - \gamma)$. Contrarily, for $p \to 1$ or $m \to 0$, it goes to its maximal value of $1/(1 - \gamma)$.*

## 4 The Augmentation Approach

In this section, we consider state-augmentation for solving MDPs with execution delay. We begin with defining an equivalent MDP with a larger state space that memorizes all missing information for an informed decision. Due to the full observability, the resulting optimal augmented policy attains the optimal return in the original delayed MDP.

---

[1]The exact terminology used by Katsikopoulos & Engelbrecht (2003) is *action delay*, while in (Bertsekas et al., 1995)[Section 1.4] it is *time lag*. We prefer the term *execution delay* since the action is itself decided instantaneously.

**Definition 4.1** (m-AMDP). *Given MDP $(\mathcal{S}, \mathcal{A}, P, r, \gamma)$ and $m \in \mathbb{N}$, an m-Augmented MDP (m-AMDP) is a tuple $(\mathcal{X}_m, \mathcal{A}, F, g, \gamma)$ such that $\mathcal{X}_m := \mathcal{S} \times \mathcal{A}^m$ is the augmented state-space, $\mathcal{A}$ the original action-space, $F$ is the transition matrix given in Appx. B.1 (14), and g is the reward function given in Appx. B.1 (15).*

The pending action queue is concatenated to the original state to form an augmented state $x_t := (s_t, a_t^{-1}, \cdots, a_t^{-m}) \in \mathcal{X}_m$, where $a_t^{-i}$ is the $i$-th pending action at time $t$. It means that in the following step, $t+1$, action $a_t^{-m}$ will be executed independently of the present action selection, the queue will shift to the right, and the newly selected action will be at the second coordinate. By construction, the $m$-AMDP is non-delayed; it directly accounts for execution delay through its state-representation, as opposed to our coming formulation in Sec. 5. We further define a stationary deterministic policy $\bar{\pi} \in \bar{\Pi}_m^{\text{SD}}$ with corresponding decision rule $\bar{d} : \mathcal{X}_m \to \Delta_{\mathcal{A}}$ and *augmented value function* $v^{\bar{\pi}}(x) := \mathbb{E}^{\bar{\pi}} \left[ \sum_{t=0}^{\infty} \gamma^t g(\tilde{x}_t, \tilde{a}_t) | \tilde{x}_0 = x \right]$. As in (1), our goal is to solve $\bar{v}^*(x) = \max_{\bar{\pi} \in \bar{\Pi}_m^{\text{SD}}} v^{\bar{\pi}}(x), \quad \forall x \in \mathcal{X}_m.$

We now analyze the classical Policy Iteration (PI) algorithm (Howard, 1960) for $m$-augmented MDPs and provide a finite-time analysis of its convergence. We refer to it as $m$A-PI and provide its pseudo-code in Appx. B.2. We consider PI since it is a canonical representative upon which many other algorithms are built. Admittedly, we did not find any other formal result quantifying the effect of augmentation on a planning or learning algorithm, other than a PAC upper bound for R-max with $\epsilon$-optimal policies (Walsh et al., 2009). A proof for the next result is given in Appx. B.4.

**Theorem 4.1** (Lower Bound for $m$A-PI). *The number of iterations required for $m$A-PI to converge in m-AMDP $\mathcal{M}_m$ is $\Omega(|\mathcal{X}_m|) = \Omega(|\mathcal{S}||\mathcal{A}|^m)$.*

Thm. 4.1 does not take advantage of the special delay problem structure but rather is an application of our more general result to augmented MDPs (Appx.B.4). As pointed out in Scherrer et al. (2016), the lower-bound complexity of PI is considered an open problem, at least in the most general MDP formulation. Lower-bounds have been derived in specific cases only, such as deterministic MDPs (Hansen & Zwick, 2010), total reward criterion (Fearnley, 2010) or high discount factor (Hollanders et al., 2012). Even though we did not intend to directly address this open question, our lower bound result seems to be a contribution on its own to the general theory of non-delayed MDPs.

Next, we show that the above lower bound is tight (up to a factor of $|\mathcal{A}|$ and logarithmic terms) and $m$A-PI is guaranteed to converge after $\tilde{O}(|\mathcal{S}||\mathcal{A}|^{m+1})$. A proof is given in Appx. B.5.

**Theorem 4.2** (m-PI Convergence). *The mA-PI algorithm converges to the optimal value-policy pair $(\bar{v}^*, \bar{\pi}^*)$ in at most $|\mathcal{S}||\mathcal{A}|^m(|\mathcal{A}| - 1) \left\lceil \log(1/\gamma)^{-1} \log(1/1-\gamma) \right\rceil$ iterations.*

## 5 EXECUTION-DELAY MDP: A NEW FORMULATION

In this section, we introduce and study the stochastic process generated by an MDP with execution delay, without resorting to state-augmentation. In the ED-MDP we consider, the probability measure changes according to the delay value $m$. We assume that during the $m$ initial steps, actions are sequentially executed according to a *fixed* queue $\bar{a} := (\bar{a}_0, \cdots, \bar{a}_{m-1}) \in \mathcal{A}^m$. Unlike $m$-AMDPs, the initial queue of pending actions here plays the role of an exogenous variable that is not embedded into the state-space. A policy $\pi \in \Pi^{\text{HR}}$ induces a probability measure $\mathbb{P}_m^{\pi}$ that is defined through a set of equations which, for brevity, we defer to Appx. C (16)-(19)]. We note that for $t < m$, decision rules do not depend on the history, while for $t \geq m$, they depend on the history up to $t - m$ only. Let $\mu$ be an initial state distribution and $\delta$ a Dirac distribution. Using this and the notations from Sec. 2, we can explicitly write the probability of a sample path. See proof in Appx. C.1.

**Proposition 5.1.** *For policy $\pi := (d_0, d_1, \cdots) \in \Pi^{\text{HR}}$, the probability of observing history $h_t := (s_0, a_0, s_1, a_1 \cdots, a_{t-1}, s_t)$ is given by:*

$$\mathbb{P}_m^{\pi}(\tilde{s}_0 = s_0, \tilde{a}_0 = a_0, \tilde{s}_1 = s_1, \tilde{a}_1 = a_1, \cdots, \tilde{a}_{t-1} = a_{t-1}, \tilde{s}_t = s_t)$$

$$= \mu(s_0) \left( \prod_{k=0}^{m-1} \delta_{\bar{a}_k}(a_k) p(s_{k+1}|s_k, a_k) \right) \left( \prod_{k=m}^{t-1} q_{d_{k-m}(h_{k-m})}(a_k) p(s_{k+1}|s_k, a_k) \right).$$

From Prop. 5.1 we deduce that, differently than the standard MDP setting where any Markov policy induces a Markov process, the delayed process is not Markovian even for stationary policies (see

Appx. C.2 for a formal proof). Next, we show that for any history-dependent policy and starting state, there exists a Markov policy (not necessarily stationary) that generates the same process distribution. Consequently, despite execution delay, one can restrict attention to Markov policies without impairing performance.

**Theorem 5.1.** *Let $\pi \in \Pi^{\text{HR}}$ be a history dependent policy. For all $s_0 \in \mathcal{S}$, there exists a Markov policy $\pi' \in \Pi^{\text{MR}}$ that yields the same process distribution as $\pi$, i.e.,*
$$\mathbb{P}_m^{\pi'}(\tilde{s}_{t-m} = s', \tilde{a}_t = a | \tilde{s}_0 = s_0) = \mathbb{P}_m^{\pi}(\tilde{s}_{t-m} = s', \tilde{a}_t = a | \tilde{s}_0 = s_0), \qquad \forall a \in \mathcal{A}, s' \in \mathcal{S}, t \geqslant m.$$

The proof is given in Appx. C.3. It builds on the concept that for each history-dependent policy $\pi \in \Pi^{\text{HR}}$, one can choose a sequence of Markov decision rules that reconstruct the same time-dependent action distribution in the process induced by $\pi$.

This result proves attainability of the optimum over $\Pi^{\text{MR}}$, but not how one can efficiently find an optimal policy. In Appx. C.5, (27), we formally define the *delayed value function* $v_m^{\mu_0:\mu_{m-1},\pi}$ for policy $\pi$ and initial action distribution queue $\mu_0 : \mu_{m-1} := (\mu_0, \ldots, \mu_{m-1})$. In Thm. C.1 there, we show that it satisfies a non-stationary Bellman-type recursion. Though the question of how to efficiently find an optimal non-stationary Markov policy remains generally open, we partially answer it by proving that a deterministic Markov policy is sufficient for the optimal delayed value function.

**Theorem 5.2.** *For any action distribution queue $\mu_0 : \mu_{m-1} := (\mu_0, \ldots, \mu_{m-1})$ and $s_0 \in \mathcal{S}$,*
$$\max_{\pi \in \Pi^{\text{MD}}} v_m^{\mu_0:\mu_{m-1},\pi} = \max_{\pi \in \Pi^{\text{MR}}} v_m^{\mu_0:\mu_{m-1},\pi}.$$

**Degradation due to stationarity.** To complement the finding that a deterministic Markov policy can be optimal for any ED-MDP, we show that restricting to stationary policies impairs performance in general. Thus, while in non-delayed MDPs it is enough to focus on the latter, in ED-MDPs the restriction should be to the more general class of Markov policies.

**Proposition 5.2.** *There exists an $m$-ED-MDP for which all stationary policies are sub-optimal.*

This result follows from computing the optimal return for stationary and non-stationary policies in the ED-MDP from Example 3.1 using simulation. We elaborate on this further in Appx. C.4. There, we also confirm that our theoretical return from Prop. 3.1 matches closely with simulation. Lastly, a visualization of the results from this section is given in Fig. 2.

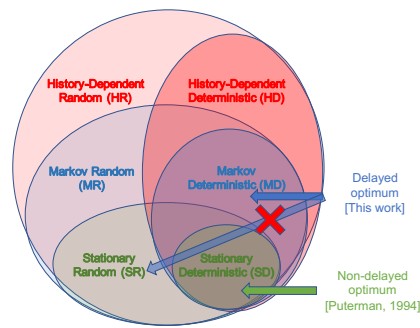

Figure 2: Optimality of policy types in ED-MDPs: Markovness is sufficient but non-stationarity is necessary.

## 6 A NEW ALGORITHM: DELAYED-Q

We now introduce an algorithm capable of successfully handling tasks with execution delay by inferring the future $m$-step state before each decision.

**Algorithm Description.** Fig. 3 depicts the algorithm. As a first stage, to select an action $a_t$ to be executed in a future state $s_{t+m}$, we infer that future state $\hat{s}_{t+m}$ using the current state $s_t$ and the queue of pending actions $(a_{t-m}, \ldots, a_{t-1})$. This is done by successively applying an approximate forward model $m$ times: $\hat{s}_{t+1} = f(s_t, a_{t-m}), \ldots, \hat{s}_{t+m} = f(\hat{s}_{t+m-1}, a_{t-1})$. More details on the forward models are given in Sec. 7. The approximate model here is simpler than other model-based algorithms such as tree-search

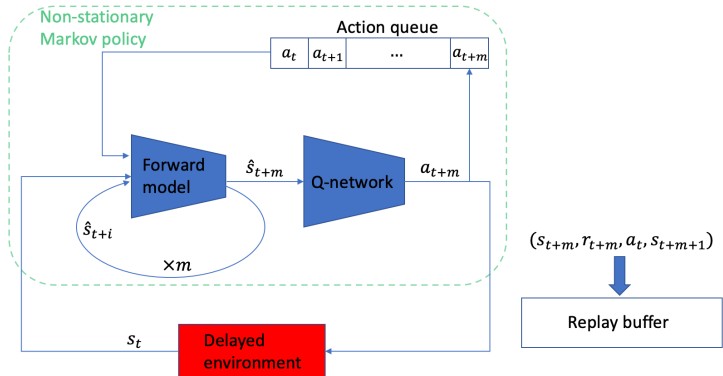

Figure 3: Delayed-Q algorithm diagram.

methods, because it does not require access to the reward function. Also, only a single trajectory is

sampled rather than exponentially many w.r.t. the horizon length. We do note this method benefits from the environment not being entirely stochastic (Walsh et al., 2009). Still, as we show next, it performs well even on noisy environments. As a second stage, we select an action according to a policy $a_t = \pi(\hat{s}_{t+m})$. The two stages of this procedure can be represented as a non-stationary Markov policy $\pi_t(s_t)$, where the non-stationarity stems from the time-dependency of the action queue, and the Markov property from the policy being applied on $s_t$ and no prior history. Notably, the Q-function here does not take past actions as input, contrarily to the augmentation approach in Sec. 4. To better stress the non-stationarity, we note that applying the policy on the same state at different times can output different actions. Lastly, for training, we maintain a sample-buffer of length $m$ which we use to shift action $a_t$ into the tuple $(s_{t+m}, r_{t+m}, a_t, s_{t+m+1})$ prior to each insertion to the replay buffer. During the course of this work, we also experimented with a model-free variant. Instead of 'un-delaying' the Q-function with the forward-model, we defined a delayed Q-function trained on sequences whose actions were shifted $m$ steps forward. However, the obtained results were unsatisfactory, seemingly because the Q-function is unable to implicitly learn the $m$-step transition.

**Point-Estimate Approaches.** For completeness, we mention alternatives to using a 'most-likely' state estimate, such as an expected future state. To demonstrate why point-estimate prediction can be devastating, consider an MDP where $s = (x, t)$: position and time, respectively. Starting from $s_0 = (0, 0)$, $t$ progresses deterministically, while $x$ behaves like a random walk with momentum; i.e., if $x > 0$, then $x + 1$ is more likely than $x - 1$, and vice versa. The process obviously diverges with time. Consider two actions: one is good when $|x|$ is big, and the other when $|x|$ is small. For a large delay $m$, the PDF of the state is bi-modal and symmetric around $(Z, m)$ and $(-Z, m)$ for some finite $Z$. But, a point estimate (e.g., ML or MAP) would yield a value of $(0, m)$. In addition to this example, we observe that in our Ex. 3.1, any alternative to a 'most-likely' state estimate is worse: there, the optimal policy applies actions based on the most-likely state (see proof of Prop. 3.1), while it is easy to see that any other policy weighing future state probabilities leads to lower reward.

# 7 EXPERIMENTS

We perform experiments in a wide range of domains: tabular, physical, and image-based Atari. All of them include stochasticity: In the maze we inject noise to actions; in the physical domains we perturb the masses at each step; and Atari is stochastic by nature. We compare our algorithm with two baselines: *Oblivious-Q* and *Augmented-Q*. Oblivious-Q is the standard Q-learning that ignores delay and assumes each decision to be immediately executed. Augmented-Q acts on the $m-$AMDP introduced in Def. 4.1. We test all domains on delays $m \in \{0, 5, 15, 25\}$ with 5 seeds per each run. All results are summarized in Fig. 10, and are provided in more detail with std. in Appx. D.1, Table 2.

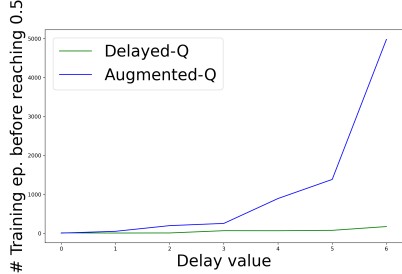

Figure 4: Maze: Time complexity as a function of $m$

**Tabular Maze Domain.** We begin with testing Delayed-Q on a Maze domain (Brockman et al., 2016)[tinyurl.com/y34tmfm9]. It is based on tabular Q-learning and enables us to study the merits of our method decoupled from the coming DDQN added complexities. Moreover, it conveys the exponential complexity of Augmented-Q. The forward-model we construct is naturally tabular as well: it predicts a state $s'$ according to the highest visitation frequency given $(s, a)$. The objective in Maze is to find the shortest path from a start position to a goal state in a randomly-generated $N \times N$ maze. Reaching the goal yields a reward of 1, and $-1/(10N^2)$ per step otherwise. The maximal episode length is $10N^2$ steps, so the cumulative reward is in $[-1, 1]$. We also create a Noisy Maze environment that perturbs each action w.p. $p \in [0, 0.5]$.

Convergence plots are given in Fig. 6. Delayed-Q outperforms the rest for all delay values $m$, while Oblivious-Q fails in all runs for $m > 0$. Since the augmented state-space grows exponentially with $m$, Augmented-Q converges more slowly as $m$ increases. In fact, for $m > 15$ the simulation fails to run due to memory incapacity for the Q-table; this explains its absence in Figs. 6–10. To confirm the exponential complexity growth of Augmented-Q and compare it with Delayed-Q, we trained both agents with increasing delay values, and reported the number of training episodes each one required before reaching a cumulative reward of 0.5. Fig. 4 clearly demonstrates the exponential (resp. linear) dependence of Augmented-Q (resp. Delayed-Q) in the delay value. The linear dependence of Delayed-Q in $m$ is not surprising: Delayed-Q is algorithmically identical to Q-learning, except

for the $m$-step forward-model calls and the replay buffer shift of $m$ samples. To further analyze its sensitivity to the state-space size, we ran tabular Delayed-Q on increasing maze sizes, for a fixed $m = 5$. As Fig. 5(c) shows, the performance drops exponentially, suggesting high sensitivity to the state-space size and highlighting one shortcoming of MBS (Walsh et al., 2009) (see Sec. 1).

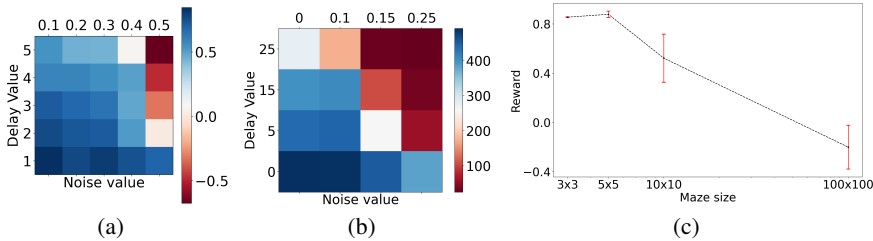

(a)                              (b)                              (c)

Figure 5: Delayed-Q (median over 5 seeds): (a) Total reward after 5000 training episodes on $10 \times 10$ Maze. Performance is sensitive to both delay value and stochasticity. (b) Noisy Cartpole. (c) Reward on varying Maze sizes. Abscissa is in log-scale, so the return decreases exponentially with $m$.

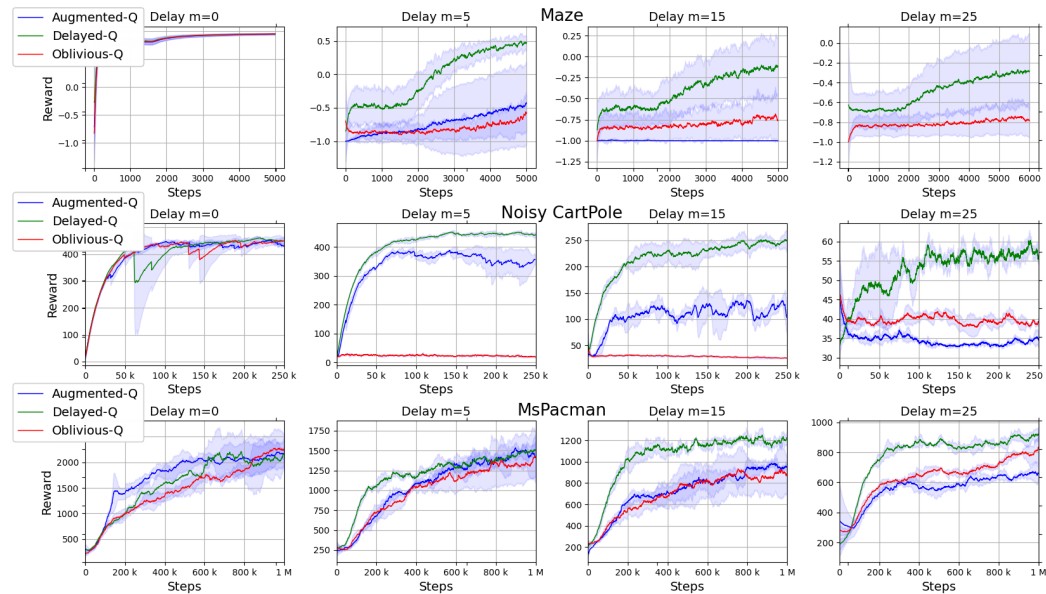

Figure 6: Convergence plots for Maze, Noisy Cartpole and Atari MsPacman. Note that the scale of the y-axes (performance) may change from figure to figure.

**Physical Domains.** Next, we test our approach on two continuous domains: CartPole[2] and Acrobot. The CartPole task requires balancing a pole connected to a cart that actuates left or right. In Acrobot, one needs to swing up the lower of two links connected by a joint above a certain height. The agent receives a reward of $1$ if the pole stays above a certain angle in Cartpole, and in Acrobot it receives $-1$ until it reaches the goal. The episode length is 500 steps in both tasks. We also create noisy versions of both tasks: At each step, normal additive noises are independently added to each physical component's mass, with std of $0.1$ of the nominal mass.

We extend the famous DDQN algorithm (Van Hasselt et al., 2015) and compare to it, though our method is general and can be seamlessly integrated into any Q-learning based algorithm. Our one-step forward-model is implemented with a neural network (NN) of the same architecture as the Q-network. Namely, it consists of two hidden layers, each of width 24, with ReLu activations. The input of the

---

[2]Since Cartpole fails in $\sim 10$ steps if the initial actions are random, we initialize the $m$-lengthed action-queue with optimal actions using a pretrained non-delayed model. We wait for $2m$ steps before starting to append samples to the replay buffer to avoid unfair advantage due to these actions.

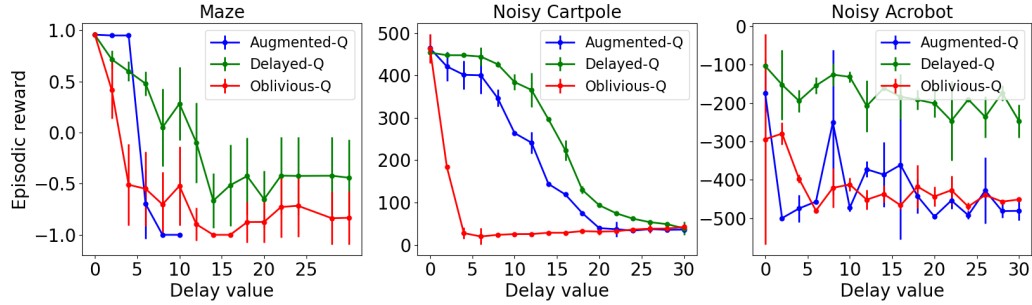

Figure 7: Performance as a function of the delay (from left to right): Maze, Noisy Cartpole, Noisy Acrobot. For Augmented-Q in Maze, $m > 10$ is missing due to explosion of the state-space.

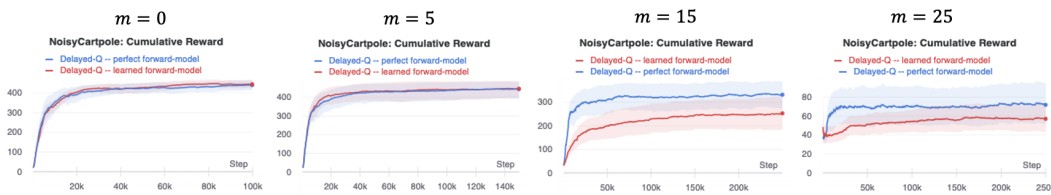

Figure 9: Noisy Cartpole: Performance gap between true and trained forward model.

forward-model NN is the concatenation of $(s, a)$ and its output is $s'$. Training the forward-model NN is conducted together with the Q-network training with the same hyperparameters and sample batches; this makes the implementation easy and simple. For Augmented-Q, a concatenation of the pending actions to the state is fed to the Q-network.

Fig. 6 depicts the performance of the three algorithms for different values of $m$ for Noisy Cartpole. As expected from a physical domain, ignoring delay gives catastrophic results even for $m = 5$. Augmented-Q performs moderately up to $m = 15$, but fails for larger delays. Delayed-Q performs the best for all $m$ values, and performs well even on the challenging task of balancing a noisy pole with $m = 25$. We observe similar behavior in all Cartpole and Acrobot experiments, as shown in Fig. 10. Moreover, in Fig. 7, we demonstrate the relative robustness of Delayed-Q to different delay values. All tested environments exhibit superior performance of Delayed-Q for a wide range of delays. In Noisy Acrobot, Delayed-Q performs better for $m = 25$ than the alternatives do for $m = 2$. Figs. 5(a)-5(b) show a clear trade-off between noise and delay, as we also discuss in Rmk. 3.1. For high

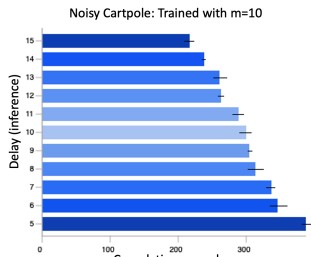

Figure 8: Performance gap for Delayed-Q trained with a delay of $m = 10$.

delays, the agent is much more sensitive to an increase in stochasticity.

To quantify the dependence of Delayed-Q on the model accuracy, we compare the learned model to a perfect one, i.e., the environment itself. Fig. 9 shows performance is impaired more as the delay increases and suggests a better model can potentially improve reward by 20-30%. Further, we test the robustness of Delayed-Q to misspecified delay by training it with $m = 10$ and evaluating on other delay values. Fig. 8 shows the evaluation performance for $m \in \{5, \ldots, 15\}$. It demonstrates the robustness of our method – varying performance in evaluation (for good or bad) does not stem from delay misspecification. Instead, the delay is 'forgotten' after training, and Fig. 8 depicts the general effect of execution delay on performance. For shorter delay than the training one, i.e., $m < 10$, performance even improves. The reason is that, first, during training, the Q-function is 'un-delayed' due to the replay buffer shift that relates the actions to the correct execution time. Second, the forward-model is trained based on single-step transitions and only during inference is it queried $m$ times. Thus, these two networks composing the agent are oblivious to the delay they were trained on.

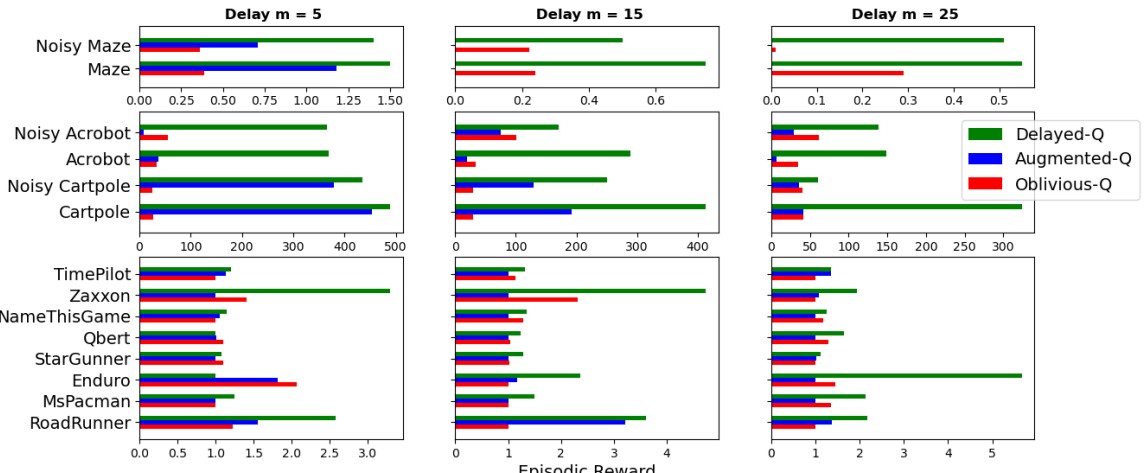

Figure 10: Experiment summary: mean of episodic return for all domains. Delayed-Q outperforms the alternatives in 39 of 42 experiments. Due to negative reward, a positive translation of 1 is applied for Maze and 500 for Acrobot. Atari x-axis is the gain relative to lowest result in each experiment.

**Atari Domains.** We run the last set of experiments on the Atari Learning Environment (Bellemare et al., 2013). We inspect 8 games from those that were successfully tackled with the original Q-network architecture and hyperparameters of DDQN (Van Hasselt et al., 2015). Since a learned forward-model for images conditioned on actions is a hanging question in the research frontier, we leave it for future work and use the simulator itself for prediction. It is stochastic in nature and thus encompasses approximation error. For Augmented-Q, we concatenate the action queue to the output of the CNN part of the Q-network; the extended vector is then fed into the subsequent fully-connected part of it. We train all games for 1M steps. Fig. 6 shows convergence plots for MsPacman. Delayed-Q is consistently better than Augmented-Q for all $m$ values, which is, in turn, better than Oblivious-Q. Although the gap between all three algorithms is small for $m = 5$, it increases with $m$. For $m = 25$, the delay is too large for the augmentation to have a positive effect compared to Oblivious-Q, and they perform the same. This behavior is representative of all Atari games, as can be seen in Fig. 10. Lastly, we compared Delayed-Q with a fourth algorithm which uses an RNN policy that is unaware of the delay value. The results are given in Appx. D.2, showing that a recurrent policy does not improve upon Augmented-Q or Oblivious-Q. This result is not surprising though: as stated in Thm. 5.1, the history sequence $s_{t-m}, s_{t-m-1}, \ldots$ does not aid the policy any further than only using $s_{t-m}$.

## 8 DISCUSSION

In this work, we found that non-stationary deterministic Markov policies are optimal in delayed MDPs. Though more expressive, the standard state augmentation approach is intractable for all but the shortest delays, while the oblivious approach that ignores delay suffers from inferior performance. We derived a Q-learning based algorithm that generates a Markov policy by combining a transition forward model with Q-network. The forward-model produces a simple future-state estimate. Incorporating probabilistic estimates and other improvements such as integration of image-based action-dependent learned forward-models (Kim et al., 2020), are left for future research. Extensions of our work for real-world applications can be unknown or varying delay. In the first case, a good prior for the delay value can often be used, e. g., for autonomous vehicles, as the latency statistics of the different hardware and software components are well studied (Zhao et al., 2019; Niu et al., 2019), while in production systems, they are almost constant (Toschi et al., 2019). Our algorithm is also readily extendable to the second case of varying delay. Differently from the augmentation approach, our 1-step forward-model decouples the algorithm from the delay used for training, as Fig. 8 depicts. Also, quantization of the delay is not essential as long as the forward model can operate with variable delay values. Finally, our framework can be extended to policy-gradient-based methods that are particularly useful for continuous control, where observation delay is inherent.

ACKNOWLEDGEMENTS

The authors would like to thank Daniel J. Mankowitz and Timothy A. Mann for motivating this work.

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
