# OpenReview forum: "Acting in Delayed Environments with Non-Stationary Markov Policies"
_ICLR.cc/2021/Conference — ICLR 2021 Poster_

### Official Review · AnonReviewer4 · 2020-10-27
**This paper is a solid study of the problem with lots of potential research directions for the future work**

**Rating:** 8
**Confidence:** 4

**Review:**

The paper presents theoretical analysis of MDPs with execution delay together with an algorithm that achieves better performance on the task than the baselines. The main theoretical result highlights the need for non-stationary Markov policies that is different from standard MDPs that can be solved using stationary Markov policies.

*Quality*
The authors conducted a solid theoretical study of MDPs with the execution delay. The presented claims showcase why the existing approach based on augmenting the state space is not feasible for large delays. The suggested algorithm is a based on a simple idea to estimate the state of MDP m steps in the future, but it seems to work quite well when the MDP is not too stochastic. Overall, this paper is a solid study of the problem with lots of potential research directions for the future work.

*Clarity*
The paper is well-written in general.

*Originality*
To my best knowledge, the results are new and the need for non-stationary policies is a novel highlight.

*Significance*
rather significant, execution delay is a common issue in practice and the paper lays foundations for analysis of MDPs with execution delay.

Pros
* Theoretical analysis of ED-MDPs that guides the presented algorithm
* Great results on Tabular Maze and Physical domain problems

Cons
* No analysis on how the stochasticity of environment affects the performance of Delayed-Q
* Atari results use the simulator to predict the future state

---

> ### Author Response · Authors · 2020-11-16
> **Response to Reviewer 4**
>
> We wish to thank the reviewer for his time and attention to detail.
>
> The effect of increasing stochasticity on the performance of delayed-Q can be seen in the heatmaps in Figs. 3(a)-(b): If we look at both figures according to the horizontal axis, we can see the effect of noise on the reward in Delayed-Q. As seen there, this negative impact intensifies as the delay increases.
>
> Regarding the simulator used for Atari, we agree that combining a learned model is a worthy topic of future research, as noted in Sec. 7. A recent work worth noting in this context is [10], which can potentially serve as such a learned model.
>
> **References**
> [10] Kim, S. W., Zhou, Y., Philion, J., Torralba, A., & Fidler, S. (2020). Learning to Simulate Dynamic Environments with GameGAN. In Proceedings of the IEEE/CVF Conference on Computer Vision and Pattern Recognition (pp. 1231-1240).

---

### Official Review · AnonReviewer3 · 2020-10-28
**A learning-based approach to solving MDPs with control or observation delays**

**Rating:** 6
**Confidence:** 4

**Review:**

This paper reminds us of the problem of delayed interaction (in control or observation) with MDPs, provides two theoretical formulations of the problem, and then a deep RL-based approach.

This is an interesting problem that has not been much talked about in the recent literature as far as I know.  It seems plausible that it would come up in real situations although the paper only shows us artificially delayed versions of artificial problems.  The paper would be much stronger if it were to address some real applications (in which, for example, we could see whether the approximate predictive model works well enough).

The theoretical analysis of the augmentation approach seems reasonable (but I did not check the math).  However, since the m-AMDP is another MDP, could we not just use existing computation-time analysis of PI (or VI or LP-solving) applied to the m-AMDP?  Or, if this bound is smaller (because it's taking advantage of special structure in the mAMDP), that's cool, and it should be clearly pointed out.

I wasn't quite clear on the contribution of the execution-delay MDP formulation.  It was interesting, but didn't seem to lead to any new algorithmic approaches (and in fact seems to present some substantial algorithmic challenges) and it isn't obvious to me what makes it better than the augmented formulation.   If it was intended to be a lead-in to Delayed-Q, then I didn't quite see the connection and it would be good to make it clearer.

The delayed-Q algorithm seems sensible.   I didn't understand the phrase "this method relies on the environment not being entirely stochastic".   Perhaps this is an allusion to the fact that it's using a determinized "point" prediction of the next state?  Although the paper makes it clear that this is an approximation, it would have been nice to have a concrete example of a kind of MDP where this is a particularly poor strategy (e.g., returning an "average" next state, which is actually impossible).

As I understand it, the delayed-Q method is relatively similar to MBS: more efficient (because it doesn't consider multiple state sequences), but potentially less accurate.

In summary, I would say the main algorithm is reasonable but perhaps not highly novel, and that the theoretical work in sections 4 and 5 does not seem to make a substantial contribution.  I think a promising direction would be to push toward some real application domains and let their demands drive the architectural and algorithmic improvements you need to address them.

=== Update ===
The arguments in the rebuttal clarify some confusions I had and illuminate the contributions of the paper further.   I am now completely on the fence about acceptance, but will tip toward positive.

---

> ### Author Response · Authors · 2020-11-16
> **Response to Reviewer 3 -- Part 1/2**
>
> We thank the reviewer for the time spent on reading the paper and for his attention to detail. We shall address his concerns in order and integrate the relevant parts into the revised version.
>
> ***Address real applications:***
> Generally, this work is principled and includes a substantial amount of theory along with multiple experiments on domains that are common in RL literature. Nonetheless, we agree it can be naturally extended to real-world applications. In such extensions, aspects that are likely to be important are varying and/or unknown values. As noted in Sec. 8 (Discussion), we believe our method can cope with varying delays thanks to the single-step-based forward-model. As for unknown delay, one could incorporate an estimation of it, which may even depend on the state. Yet, as we pointed out to Reviewer 2, many real-world systems can be reduced to a problem with known delay. Also, when the delay is misspecified during training, our agent does not suffer from degradation, as shown in the new experiment we added: https://www.dropbox.com/s/87msatlmgnqeuvd/robustness.png?dl=0 -- training on $m=10$ and evaluating on other delays; see more in response to Reviewer 2. We recall that working with a delay in RL and control, in general, is considered one of the hurdles for real-world RL; see [4]. We believe that our work will open up avenues for applied research.
>
>
> ***Use existing computation-time analysis of PI to the m-AMDP:***
> It is correct that existing results on PI hold for m-AMDPs, since these are regular MDPs. However, as pointed out in [5], the lower-bound complexity of PI is still an open problem, at least in the most general MDP formulation. Actually, lower-bounds have been derived in specific cases such as deterministic MDPs [6], total reward criterion [7] or close to 1 discount factor [8]. Even though we did not intend to directly address this open question, the literature survey we conducted for this rebuttal suggests that our result is a contribution on its own to the general theory on MDPs (without delay). We stress that, in fact, we do not take advantage of the special delay problem structure (see Appx. B.4, first Prop.). We thank the reviewer for this observation and shall emphasize this contribution.
>
>
> ***Contribution of the execution-delay MDP formulation; substantial algorithmic challenges; comparison with the augmented formulation; lead-in to Delayed-Q:***
> First, our intention is to claim that albeit augmented MDPs is “the obvious thing to do”, it encompasses several disadvantages. We highlight them both theoretically (Prop. 4.1 and Thm. 4.1) and empirically (Fig. 6 and others). This motivates the study of delay in non-augmented MDPs through execution-delay MDPs.
>
> Our first contribution (Thm 5.1) is in showing that limiting the most general class of history-dependent policies to Markov policies does not impair performance. We believe this is in fact dramatic: while non-delayed RL relies on the ability to restrict policy search to the space of Markov policies, we show for the first time that this applies to the delayed case as well. Our second contribution (Prop. 5.2) is in showing that stationary policies are suboptimal in general. We believe this is again significant: while for non-delayed MDPs one can restrict policy search to stationary policies, we show that in the delayed case, this yields suboptimal performance. These two results, therefore, lead to focusing attention on Markov non-stationary policies, and provide the theoretical justification to our Delayed-Q algorithm, which indeed uses Markov non-stationary policies in the original, non-augmented state-space.
>
> Therefore, regarding the reviewer’s concern, while the derivation of Delayed-Q indeed does not rely on the delayed Bellman equations (Appx. C.5), it does match the theoretical insights on execution-delay MDPs.
>
> ***References***
> [4] Dulac-Arnold, G., Mankowitz, D., & Hester, T. (2019). Challenges of real-world reinforcement learning. arXiv preprint arXiv:1904.12901.
>
> [5] Scherrer, B. (2016). Improved and generalized upper bounds on the complexity of policy iteration. Mathematics of Operations Research, 41(3), 758-774.
>
> [6] Hansen, T. D., & Zwick, U. (2010, December). Lower bounds for Howard’s algorithm for finding minimum mean-cost cycles. In International Symposium on Algorithms and Computation (pp. 415-426). Springer, Berlin, Heidelberg.
>
> [7] Fearnley, J. (2010, July). Exponential lower bounds for policy iteration. In International Colloquium on Automata, Languages, and Programming (pp. 551-562). Springer, Berlin, Heidelberg.
>
> [8] Hollanders, R., Delvenne, J. C., & Jungers, R. M. (2012, December). The complexity of policy iteration is exponential for discounted Markov decision processes. In 2012 IEEE 51st IEEE Conference on Decision and Control (CDC) (pp. 5997-6002). IEEE.

---

> > ### Author Response · Authors · 2020-11-16
> > **Response to Reviewer 3 -- Part 2/2**
> >
> > ***A concrete example of an MDP where point estimate is a poor strategy:***
> > The reviewer raised an important point; here is an example to demonstrate why point-estimate prediction can be devastating. Consider an MDP where two parameters define the state $s=(x,t)$. Starting from $(0,0)$, the second index, time, progresses deterministically, while the first index behaves like a random walk with momentum. I.e., if $x>0$, $x+1$ is more likely than $x-1$, while if $x<0$, $x-1$ is more likely than $x+1$. The process obviously diverges with time.
> > Now, suppose there are two actions: one is good in extreme values (when $|x|$ is big) and one is good in values close to $0$ (when $|x|$ is small). Suppose also that the delay $m$ is large. The PDF of the state is bi-modal and symmetric around $(Z,m)$ and $(-Z,m)$. But, the point estimate (e.g., ML, or MAP) will give a value of $(0,m)$.
> >
> > ***Similarity to MBS:***
> > MBS is a conceptual algorithm, so we find the comparison to MBS a bit problematic in general. Evidently, even with the model approximation considered there, MBS cannot run on domains like Atari, as planning is performed on a discretized space. Differently, Delayed-Q works with the original, possibly continuous state-space. Moreover, MBS is an offline algorithm: it estimates a surrogate, non-delayed MDP from samples, and only then, as a final stage, solves that MDP to obtain the optimal policy (see [9], Algorithm 2, line 16). This is inapplicable to large continuous domains and is in contrast to Delayed-Q.
> >
> > **References**
> > [9] Walsh, T. J., Nouri, A., Li, L., & Littman, M. L. (2009). Learning and planning in environments with delayed feedback. Autonomous Agents and Multi-Agent Systems, 18(1), 83.

---

### Official Review · AnonReviewer1 · 2020-10-31
**well-motivated study**

**Rating:** 6
**Confidence:** 4

**Review:**

This paper investigated the problem of learning agents when there are execution delays. The authors (i) used a two-state MDP example to show some equivalence between execution delay and stochasticity of transitions; (ii) analyzed the action aggregation method, which cumulated all the history and then made decisions. They show a classic Policy Iteration (PI) method with the aggregation, unfortunately, has its iteration complexity exponentially depending on the delay time $m > 0$; (iii) formulated the Execution-Delay (ED)-MDP and showed that there exists a non-stationary Markov policy which attains optimal value, while any stationary policies will have suboptimal performance; (iv) proposed a model-based Q-learning method, delayed-Q, which used the predicted future state-action sequence to make decisions; (v) did experiments on Maze, CartPole, Acrobot and Atari tasks to verify the proposed delayed-Q method.

Pros:
1. The execution delay seems to be an important problem in practice. The motivation is convincing.
2. The theoretical study is fairly strong and insightful.
3. The experiments show that the proposed method promisingly works well.

Questions:
1. The equivalence between execution delay and stochasticity of transitions is interesting. If we use usual assumptions (immediate action execution, and stochastic transitions), does this observation implies inherently there is still something equivalent to "execution delay"?
2. The PI method is proved to have exponential dependence on $m$. Is the proposed delayed-Q learning method guaranteed to have much nicer dependence on $m$?
3. The proposed delayed-Q is model-based (although as noted it does not need a model of transitions), any thought about how to adapt it into model-free settings?
4. If the environment is stochastic, then the proposed method will be largely impacted since the next state will be much harder to estimate. Any reason or evidence for if / why this method is expected to perform well?
5. Also, it seems the comparison is unfair in terms of model-based delayed-Q vs. model-free Augmented Q-learning. How much benefit does the delayed-Q gain from the learned model (or how sensitive the delayed-Q is w.r.t. the model error)? It would be great if there is some investigation over this issue.

Overall, I found the problem is important, and the motivation is convincing. The theoretical study of the execution delay problem is systematic and solid. The proposed method is promising and verified by experiments. On the other hand, there are still a number of questions that need to be clarified, such as the theoretical guarantees of the proposed delayed-Q, and issues about its model-based nature.

===Update===

After reading the authors' feedback and other reviews, I would keep my current rating.

---

> ### Author Response · Authors · 2020-11-16
> **Response to Reviewer 1**
>
> We appreciate the time the reviewer spent on carefully reading and assessing the paper, and thank him for his insightful questions. We shall address them in order, and will integrate most of the answers in the final version.
>
> 1. We believe the reviewer refers to the tradeoff between stochasticity and delay mentioned in Remark 3.1. We agree that for a general stochastic MDP, the performance should be strictly monotonically decreasing in the delay value and in the level of stochasticity. More formally, taking $m\rightarrow \infty$ should yield the same performance as in an MDP with uniform transitions. Therefore, from a continuity argument combined with the monotonicity of the two parameters, there should be a constant performance level-curve that couples increasing delay values with decreasing stochasticity levels, and vice-versa. Our Prop. 3.1 is merely a special case of this coupling for which an exact formula is given.
>
> 2. First, we note that PI is a planning algorithm that relies on a known model while Q-learning is a sample-based algorithm that involves learning. So, the comparison we can make between them is not formal. Nonetheless, if we apply the same reasoning behind the complexity analysis of PI to Delayed-Q, we see that the latter is algorithmically identical to Q-learning, except for two parts: the $m$-step forward-model calls, and the replay buffer shift of $m$ samples. Hence, algorithmically, while Augmented-Q is exponential in $m$, Delayed-Q is in fact linear in $m$. To support this, we added the following experiment that clearly demonstrates the exponential vs. linear dependence of the two algorithms for the Maze domain: https://www.dropbox.com/s/bbqjs5gv5gwctmd/complexity_delayed_augmented.png?dl=0.
>
> 3. Excellent question, and the answer is yes. In fact, we began this research with a model-free algorithm that works as follows. Instead of the current ‘un-delaying’ we apply to the Q-function via the replay buffer shift, we defined a delayed Q-function for which the action is shifted by $m$ steps from the state, and is trained using the original state-action sequences. We trained it with two methods: online exploration and imitation of a pre-trained teacher. However, the results were unsatisfactory. We believe the reason is that in that case, the Q-function is supposed to implicitly learn the $m$-step transition (as it does with a single step in non-delayed MDPs), which is beyond its capabilities. Dedicating a forward-model for this task solved the issue.
>
> 4. The reviewer is right. Stochasticity does impact Delayed-Q since it relies on a prediction of the $m$-th next state. Nonetheless, a possible alternative could instead consider the PDF over the future $m$-th state to compute its expectation and choose the action accordingly. However, this approach can be bad, as we demonstrate with an example in our response to Reviewer 3. We also observe this phenomenon in Example 3.1: there, the optimal policy applies an action based on the most-likely state (proof of Prop. 3.1), while one can easily show that any other policy that weighs future state probabilities leads to lower reward. Lastly, our experiments demonstrate relatively high robustness to noise, as can be seen for Noisy Maze, Noisy Cartpole, and Noisy Acrobot in Figs. 3 and 6.
>
> 5. This is a good point, which continues the discussion from item 3. As explained there, an explicit model is indeed key to enabling good performance of the algorithm. As for Augmented-Q being model-free -- it is not clear how it would benefit from maintaining a model, since it converts the MDP to a non-delayed one. Also, notice that the forward-model is trained online using the same samples as the training agent, so in a sense, it comes “for free”. To answer with further investigation as requested, we added the following experiment on Noisy Cartpole. For different delay values, we measure the performance when having access to a perfect model (i.e., the environment itself) vs. the learned model. As seen in https://www.dropbox.com/s/un2dtd6nhuxysek/perfect-learned_gap.png?dl=0, model error has no impact for $m=5,$ but it becomes impactful when the delay is $m=15$ and $m=25$. Thus, for large delays, performance can potentially be improved by 20-30% with a better model.

---

### Official Review · AnonReviewer2 · 2020-11-03
**Approximate an RNN policy in non-Markov observations**

**Rating:** 5
**Confidence:** 4

**Review:**

This paper studies the problem of RL with action delay, or equivalently, non-Markov decision process.
Compared to a naive strategy of state expansion (which appends a history of actions to the observation), the authors show that amending a policy to be nonstationary can:
  (a) achieve the optimal policy in analysis for tabular MDPs
  (b) performs well in a series of computational experiments.

There are several things to like about this paper:
- The overall standard of exposition and discussion is strong, and it is easy to follow the paper.
- The paper takes on a clear issue in reinforcement learning, and comes to some equally clear recommendations and findings.
- The proposed solution method is reasonable, and appears to perform well in practice.
- The series of experiments are well thought through... with a clear progression from theory -> didactic examples -> deep RL.

However there are a few places where this paper falls short:
- I find some of the characterisations quite "toy" overall... yes it is great that the agent can perform well in the setting where m=delay is known, but there is no discussion of the robustness/tradeoff when this is not the case. In a sense this agent is given priveleged information, and while it does better than the expanded state space approach, this is still a very stylized problem.
- Related to the above, the obvious baseline would be some kind of RNN-policy, who is *not* informed of the action delay m. I would have thought this is a more practical general purpose approach, and should at least be considered and/or compared to.
- The plots in Figure 4 could be improved in a few ways... add the labels for maze/cartpole/boxing next to the actual plots. Find a way to show the dropoff with performance not just for Cartpole but also maze/boxing.

Overall I think the paper is a reasonable contribution, I'm generally a fan of these simple/clear pieces of work.
However, I feel like the current paper is missing the "elephant in the room"... state of the art agents like R2D2 are *already* using recurrent policies... that surely has to be a reasonable baseline here?
If you can include that and discuss that well then I will upgrade to an accept.

---

> ### Author Response · Authors · 2020-11-16
> **Response to Reviewer 2**
>
> We wish to thank the reviewer for carefully reading the paper and providing his valuable comments. We shall address them all and integrate the relevant parts into the revised submission:
>
> ***Discussion of the robustness/tradeoff when the delay value is unknown***
> First, we note that in many systems such as autonomous vehicles, the latency statistics of the different hardware and software components are well studied [1,2], and in production systems are almost constant [3]. In other cases, they can be estimated as the first stage.
>
> However, we agree that it is intriguing to test our algorithm on delay values that are different than those it was trained on. For that reason and to answer the reviewer’s request, we added the following experiment on Noisy Cartpole: We first trained Delayed-Q with a delay of 10. Then, we loaded both the Q-network and the forward-model to evaluate the trained agent on additional delay values it has not seen before (from 5 to 15). The results are given here: https://www.dropbox.com/s/87msatlmgnqeuvd/robustness.png?dl=0 . We believe this plot nicely demonstrates the robustness of our method -- varying performance in evaluation (for good or bad) does not stem from delay misspecification. Instead, the delay is ‘forgotten’ after training, and the figure depicts the general effect of execution delay on performance. For shorter evaluation delay than the training delay, performance even improves (m=5,6,7,8,9).
>
> The reason is that, first, during training, the Q-function is ‘un-delayed’ due to the replay buffer shift that relates the actions to the correct execution time. Second, the forward-model is trained based on single-step transitions and only during inference is it queried $m$ times. Thus, these two networks composing the agent are oblivious to the delay they were trained on.
>
> ***RNN-policy baseline***
> We agree that RNN-based policies seem like a promising approach at a first glance. However, we did not expect them to perform especially well for the following reasons. The first reason comes as a good opportunity to apply Thm. 5.1: RNN policies feed the last $N>1$ observed states to an LSTM, while according to our theorem, the sequence of states $s_{t-m}, s_{t-m-1}, \dots$ does not aid the policy compared to only using $s_{t-m}$. So a naive RNN-policy that takes the last $N$ observed states is not expected to have more benefit than a Markov one. An additional deficiency of an RNN-policy is that, similarly to Oblivious-Q, it targets the wrong Q-value and does not account for delayed execution. Notice that this is not the case in both Augmented-Q and Delayed-Q.
>
> Despite the reasons above, we agree a comparison to an RNN-based policy is still a relevant baseline, and to answer the reviewer’s request, we ran such an experiment on Atari. On a technical note, we found implementations of R2D2 in other repositories than ours, which prevents a direct comparison with our algorithm. Also, R2D2 is a distributed algorithm, which adds another level of complexity. The repository we base our implementation on is the well-known “stable-baselines”. The RNN policy there is A2C, so we choose to work with that. A game where we found that A2C does not diverge is Atari’s “Frostbite”. The results are shown here: https://www.dropbox.com/s/if95h1zq9hd3lfn/lstm_comparison.png?dl=0. As seen, using a recurrent policy does not seem to improve upon Augmented-Q or Oblivious-Q.
>
> ***Improve plots in Figure 4***
> We thank the reviewer for the advice. We will add the labels as requested. Additionally, as per the reviewer’s request, we added two delay-sensitivity experiments on Maze and Noisy Acrobot, as originally done in Fig. 5 (it will take a significant amount of time to run such a sweep on Atari): https://www.dropbox.com/s/qaodb66nhwmak7t/degradation_maze.png?dl=0, https://www.dropbox.com/s/qw4ny3svj0epglp/degradation_acrobot.png?dl=0 (notice that for Augmented-Q in Maze, experiments with $m>6$ are missing due to explosion of the state-space). Both experiments again exhibit superior performance of Delayed-Q compared to Oblivious-Q and Augmented-Q, for a wide range of delays.
>
>
> **References**
> [1] Zhao, H., Zhang, Y., Meng, P., Shi, H., Li, L. E., Lou, T., & Zhao, J. (2019). Towards Safety-Aware Computing System Design in Autonomous Vehicles. arXiv preprint arXiv:1905.08453.
>
> [2] Niu, W., Ma, X., Wang, Y., & Ren, B. (2019). 26ms inference time for Resnet-50: Towards real-time execution of all DNNs on smartphone. arXiv preprint arXiv:1905.00571.
>
> [3] Toschi, A., Sanic, M., Leng, J., Chen, Q., Wang, C., & Guo, M. (2019, August). Characterizing perception module performance and robustness in production-scale autonomous driving system. In IFIP International Conference on Network and Parallel Computing (pp. 235-247). Springer, Cham.

---

### Author Response · Authors · 2020-11-25
**Rebuttal Revision Updated**

We again wish to thank the reviewers for their valuable feedback. We uploaded a revised version that incorporates our answers to all the questions raised by the reviewers. Particularly, the fruitful reviews yielded five new experiments in Figures 3, 6, 7, 8, 11, and multiple text clarifications. To preserve anonymity, we shall add a link to our code upon acceptance.

---

### Decision · Program_Chairs · 2021-01-07
**Final Decision**

**Decision:**

Accept (Poster)

**Comment:**

The paper analyzes MDPs with execution delays. Interesting theoretical results and experiments are provided, which show the benefits of the proposed algorithms. However, some issues are highlighted in the reviews, such as the lack of theoretical analysis of the proposed delayed Q-learning method, and the simplicity of the experiments. The latter is at least partially addressed by the authors in the rebuttal, and the new experiments should be incorporated in the final paper.